# Automated Damage Detection Using Lamb Wave-Based Phase-Sensitive OTDR and Support Vector Machines

**DOI:** 10.3390/s23031099

**Published:** 2023-01-18

**Authors:** Rizwan Zahoor, Ester Catalano, Raffaele Vallifuoco, Luigi Zeni, Aldo Minardo

**Affiliations:** Department of Engineering, Università della Campania Luigi Vanvitelli, Via Roma 29, 81031 Aversa, Italy

**Keywords:** distributed optical fiber sensors, structural health monitoring, Lamb waves

## Abstract

In this paper, we propose and demonstrate a damage detection technique based on the automatic classification of the Lamb wave signals acquired on a metallic plate. In the reported experiments, Lamb waves are excited in an aluminum plate through a piezoelectric transducer glued onto the monitored structure. The response of the monitored structure is detected through a high-resolution phase-sensitive optical time-domain reflectometer (ϕ-OTDR). The presence and location of a small perturbation, induced by placing a lumped mass of 5 g on the plate, are determined by processing the optical fiber sensor data through support vector machine (SVM) classifiers trained with experimental data. The results show that the proposed method takes full advantage of the multipoint sensing nature of the ϕ-OTDR technology, resulting in accurate damage detection and localization.

## 1. Introduction

In recent years, structural health monitoring (SHM) methodologies based on guided ultrasonic waves has attracted a great deal of interest, as they permit covering relatively large areas with a small number of sensors [1]. Lamb wave methods are sensitive to small damage, e.g., fatigue cracks in metallic structures or disbands and delamination in composite structures [2]. Lamb wave-based damage detection techniques rely on the excitation of ultrasound-guided waves into some point of the structure, which are sensed in a different location. The detected wave is influenced by any discontinuity (e.g., crack) occurring along the propagation path. Therefore, comparing the material response with the baseline measurements, one may detect or even classify the severity of the damage [3].

While Lamb wave signals are efficiently detected through piezoelectric transducers, many of these sensors are required to ensure good coverage and damage detection ability in complex structures [4]. This leads to increased cost and complexity of the SHM system. As an alternative, optical fiber sensors such as fiber Bragg gratings are lightweight and smaller than the piezoelectric ones. Furthermore, several sensors can be multiplexed along a unique optical fiber, which can be interrogated at a high acquisition speed [5]. However, even for FBGs, the cost increases with the number of installed sensors. Furthermore, there is a limit on the maximum number of sensors that can be multiplexed along the same fiber. Recently, we demonstrated that using the phase-sensitive optical time-domain reflectometry (ϕ-OTDR) [6], a single optical fiber can detect Lamb waves in distinct regions of the fiber. As with the multiplexed FBG sensing technology, the ϕ-OTDR method can potentially replace conventional SHM sensors such as strain gauges and PZT transducers, with the advantage of offering several measurement points along the same fiber. More precisely, the number of equivalent sensing points will depend on the ratio between the length of the sensing fiber and the sensor’s spatial resolution. Therefore, the larger the number of equivalent sensing points, the more convenient is the use of the ϕ-OTDR method (as with any other distributed sensing technology).

In general, damage detection and characterization using Lamb waves is a difficult task, as even simple structural configurations can lead to complex response signals [7]. Furthermore, the amplitude of the detected signal is often affected by changes in the coupling conditions between the transducer and the structure [8]. Even worse, the excitation of high-order Lamb modes at boundaries and discontinuities leads to complex response signals that are difficult to analyze and interpret. Given the difficulty in obtaining an accurate physical modeling of the Lamb wave interaction with damages in metallic and composite structures, several machine learning (ML) approaches have been proposed aimed at diagnosing the structural condition from measured data without the need for physics-based structural models. Among these, supervised ML approaches develop a classifier based on the availability of training data that can assign a class label to unseen data. In the context of SHM, this class label could indicate the location or severity of damage or simply indicate whether the structure is damaged or not. In [8], a supervised learning regressor based on artificial neural networks (ANNs) was developed, providing real-time damage detection in an aluminum plate equipped with six piezoelectric sensors. In [9], a multiheaded 1-D convolutional neural network able to operate directly on raw time-domain Lamb wave signals recorded from a thin metallic plate was developed to perform damage detection. Support vector machines (SVMs) have been proposed as well, either for the detection and classification of defects [10,11] or for temperature compensation of the response signals [12].

Independently of the adopted ML algorithm, the availability of a sufficiently large dataset is essential to achieve good performance, both in terms of accuracy and generalization capability. The use of distributed optical fiber sensing technologies offers the opportunity to gather many signals from the monitored structure using a single fiber. The combination of ML methodologies and distributed optical fiber sensing has been proposed e.g., in [13], where a five-layer convolutional neural network (CNN) was used to classify six seismic events in the vicinity of the sensing fiber. In [14], an artificial neural network trained with simulated data was used to detect and classify acoustic events, such as footsteps and vehicles. In [15], different ML techniques (including ANNs and SVMs) were applied to the characterization of a fluid flow in pipes. A comprehensive review on the integration of ML algorithms with distributed optical fiber sensor technologies can be found in [16].

In this paper, we take full advantage of the distributed nature of the proposed sensor by training SVM classifiers with Lamb wave signals captured in different positions along a sensing fiber. Of the generated dataset, 75% was used for SVM training, while the remaining 25% was used for testing the performance of the trained models. The experimental results show that the proposed approach is able to reveal the presence of a defect simulated by a lumped mass of 5 g, and even localize the defect with good accuracy.

## 2. Phase-Sensitive Optical Time-Domain Reflectometry

The configuration used to excite and capture the Lamb waves in a thin metallic plate is depicted in Figure 1. The test plate is an aluminum plate with in-plane dimensions of 300 × 500 mm and thickness 2 mm, and kept in resting conditions. The Lamb waves were generated using a piezoceramic actuator disk with diameter of 10 mm and thickness of 0.25 mm glued onto the plate surface using cyanoacrylate adhesive. Figure 1 also shows the positions D1, D2, D3 and D4 chosen for the placement of the 5 g mass simulating the defect. The added mass, in the form of a cylinder with a diameter of 12 mm and a height of 8 mm, was simply placed on the upper surface of the plate. The excitation signal applied to PZT for the generation of the Lamb wave was formed by a five-cycle sine wave at 38 kHz with a Hamming window. The signal was produced using an arbitrary waveform generator connected to two linear amplifier modules (MX200, PiezoDrive, Shortland, Australia) amplifying the signal and driving the PZT up to a peak-to-peak voltage of 400 V. Note that without driving signal amplification, the resulting Lamb wave would be too weak to be accurately detected by our optical fiber sensor.

The ultrasound vibrations produced on the plate by the actuator were captured by an optical fiber glued on the plate using cyanoacrylate adhesive (see Figure 1). A specialty optical fiber (AcoustiSens by OFS), with a distributed weak Bragg grating inscribed along its entire length and featuring a backscattered Rayleigh signal enhanced by 13 dB compared to a conventional single-mode fiber was used for the experiments. The fiber was glued on the plate along two parallel lines (OFS1 and OFS2 in Figure 1) separated by 150 mm. A fiber loop of 3 m was left between the two fiber strands, with the aim of separating the corresponding backscattered signals.

The homodyne configuration [17] depicted in Figure 2 was used for phase-sensitive OTDR measurements. In brief, the output of a narrowband (<7 Hz) laser (OE4028, OEwaves, Pasadena, CA, USA), operating at 1550 nm, is sent into two fiber branches using a 90/10 optical coupler, so creating two signals: one acting as the optical local oscillator (OLO) and the other one as the probe. In the probe branch, an electro-optic modulator (EOM) biased on its null point is used to produce optical pulses with a duration of 6 ns, resulting in a spatial resolution of 60 cm. After amplification through an erbium-doped fiber amplifier (EDFA100P, Thorlabs, Newton, NJ, USA), the probe pulses are launched into the sensing fiber through an optical circulator (OC). The Rayleigh backscattered light is mixed with the OLO through a 90° optical hybrid (HB-C0AFAS013, Optoplex, Fremont, CA, USA), whose outputs are sent to two identical balanced photodetectors with a bandwidth of 350 MHz (Thorlabs, PDB435C). The outputs of the two balanced photodetectors, representing the in-phase (I) and quadrature (Q) components of the backscatter Rayleigh field [17], are digitized at 2 GS/s by a high-speed digitizer (SA220P, Acqiris, Plan-les-Ouates, Switzerland), and then processed to recover both amplitude $A_{S}\left(t\right)$ and phase $\phi \left(t\right)$ of the backscatter light. The time coordinate *t* is then converted into a spatial coordinate *z,* using $z=\frac{c}{2n}t$, where c is the light velocity in the vacuum and n is the fiber’s group refractive index. Finally, the differential phase is calculated as $\Delta \phi \left(z\right)=\phi \left(z+GL\right)-\phi \left(z\right)$, with a gauge length GL = 60 cm. Compared to the heterodyne scheme used in [6], whose spatial resolution was limited to 2 m due to the bandwidth of the acousto-optic modulator, the homodyne scheme employed here allows a better spatial resolution. On the downside, the homodyne technique is slightly more complex, as it requires the use of a 90° optical hybrid and an extra balanced photodetector.

Due to the relatively short optical pulses used for the measurements, an averaging procedure was required to reach a sufficiently high signal-to-noise ratio. The procedure, already described in [6], consists of performing and averaging 320 consecutive acquisitions at a trigger rate of 10 Hz. For each trigger pulse, the 38 kHz excitation waveform was sent to the piezo-actuator, and the resulting vibration signals were acquired by performing 400 consecutive acquisitions of the ϕ-OTDR backscatter signal at a repetition rate of 450 kHz. Therefore, each acquired signal had a duration of ≈890 μs, while the total time required to perform and average 320 acquisitions at a trigger rate of 10 Hz was 32 s.

An example of the vibratory signals acquired from the OF1 and OF2 positions with the plate in pristine conditions (i.e., without addition of the lumped mass) is shown in Figure 3. While Figure 3 shows only two signals, the adopted sensor captures a vibratory signal from each position of the fiber, with a sampling step of 5 cm (as dictated by the DAQ acquisition rate). As the width of the plate is 500 mm, the fiber configuration sketched in Figure 1 provides 10 signals from OF1 and 10 signals from OF2, resulting in a total of 20 signals. Obviously, we should remark that the 10 signals received from the same glued length are somewhat correlated, as they come from positions closer than the gauge length of our sensor (60 cm).

## 3. Support Vector Machines for Damage Detection and Localization

After demonstrating that our distributed sensor is able to capture the vibratory signals generated in the plate by the piezoelectric transducer, we now discuss how to use these signals to perform a diagnosis of the investigated structure. We propose here the use of support vector machines (SVMs) for the classification of the plate conditions. An SVM classifies data by finding the best hyperplane (i.e., the one with the largest margin) that separates all data points of one class from those of the other class [18]. As with any supervised learning model, we first need to train the SVM and then cross-validate the classifier. Finally, we use the trained machine to classify (predict) new data. For our application, the signals acquired by the ϕ-OTDR in specific positions of the fiber have been used to train, validate, and test the SVM. The whole processing has been realized in MATLAB using the Statistics and Machine Learning toolbox. Differently from other works based on SVM classifiers for Lamb wave-based SHM [19,20], no specific features, such as envelope or power spectral density, were extracted from the Lamb signals. Rather, the entire signal acquired by the fiber has been used to train, validate, and test the SVMs. We further observe that the response of a structure to Lamb waves is often influenced by external factors such as temperature and load conditions [21]. Furthermore, the ϕ-OTDR signals are usually contaminated by environmental noise. Therefore, to build a robust SVM classifier, a dataset with several acquisitions of the ϕ-OTDR signal in nominally identical conditions of the plate has been generated. For our experiments, the mechanical response of the plate was recorded 200 times for each plate condition: undamaged, or with the 5 g mass added in one of the four positions depicted in Figure 1. Each dataset was split in a training dataset (75% of the acquisitions), and a test dataset (25% of the acquisitions). For each one of the five classes representing the plate conditions (undamaged or with damage D1, D2, D3, or D4), a binary SVM classifier with a quadratic kernel was built using the ϕ-OTDR signal acquired in specific positions of the fiber and providing as the output the probability (score) that the plate belongs to that class. The plate condition is then estimated based on the class providing the largest score.

## 4. Experimental Results

As a first experiment, we trained a single SVM classifier aimed only at distinguishing the pristine condition (no “damage”), from the damage condition (i.e., with the mass in any position from D1 to D4). To this aim, we used 150 acquisitions done with the plate in pristine conditions, and 150 × 4 = 600 acquisitions with the plate in the condition D1, D2, D3 or D4. In order to correctly select the fiber positions (“channels”) to be used for SVM training, we first calculated the standard deviation of the ϕ-OTDR signal acquired in each fiber position. The computed values, shown in Figure 4, represent a sort of “energy” of the vibratory signal captured at each position. Two main peaks are clearly visible in the figure, corresponding to the two fiber lengths (OF1 and OF2) glued on the plate. A third peak is also visible after OF2, partly merging with the latter, probably due to imperfect isolation of that fiber piece capturing a fraction of the plate vibration.

In order to demonstrate the capability of the proposed sensor to detect the damage condition of the plate, we built an SVM using the differential phase measured by our ϕ-OTDR sensor at z = 2.8 m, i.e., at the fiber position exhibiting the maximum vibratory energy along OF1. The whole signal was used for SVM training after normalization, resulting in 400 precursors. The SVM was cross-validated using the k-fold cross-correlation method, with k = 5 [18]. The fraction of the acquired dataset not used for SVM training, corresponding to 50 acquisitions in the “undamaged” condition, and 200 acquisitions in the “damaged” condition (50 for each damage condition D1, D2, D3 or D4) were used for SVM testing. The SVM accuracy was then calculated based on the percentage of correct predictions on the unseen test dataset. In order to demonstrate the importance of using a large dataset for SVM training, we show in Figure 5 the accuracy of the SVM tested with the same dataset as a function of the number of ϕ-OTDR acquisitions (from 50 to 750) used for training.

It is clear that by increasing the number of acquisitions, the model accuracy generally improves. Using 750 acquisitions, the SVM test accuracy is 98.40%, with a total of 0 false positives (out of 50) and 3 false negatives (out of 200).

Given the excellent prediction accuracy of the SVM model built using only one fiber channel, no relevant improvement in SVM model accuracy is expected including other fiber positions. For example, after including into the SVM the 400 precursors associated to one fiber position along OF2 (z = 6.05 m), the SVM accuracy is 97.6% (4 false positives out of 50, and 2 false negatives out of 250), thus even slightly lower than the one obtained by using only data from z = 2.80 m. The circumstance of a reduced prediction accuracy can be explained by the fact that in this case, the prevailing effect is the addition of environmental noise. In any case, we verified that any choice of the measurement channels within OF1 and OF2 always resulted in an excellent prediction accuracy (>95%).

Further tests were done with the aim of demonstrating the damage localization capabilities of the proposed methodology. To this end, 75% of the dataset acquired with the plate in its five conditions (“undamaged” or “damaged” with D1, D2, D3, and D4), composed of 750 acquisitions, was used to train five SVM binary classifiers. Each SVM classifier provides, as a score, the probability that the plate belongs to that class. After training, testing was done using the remaining 25% of the dataset, composed of 250 acquisitions. In this case, an array of five scores was obtained for each acquisition. The predicted class is the one resulting in the highest score. At first, the five SVMs were trained using only the optical signal acquired at the channel z = 2.80 m. The testing results are reported in Table 1 (first row). The table shows in its first column the fiber channel used for SVM training, while the second column shows the testing accuracy, calculated as the percentage of correct predictions over the whole testing dataset. The remaining columns show the number of correctly predicted observations relative to each class.

The table indicates that by using only the channel z = 2.80 m, the overall accuracy is 83.60%. With the aim of improving the performance, we progressively added new channels for SVM training and testing. For example, from Table 1 we see that by adding the position z = 6.05 m (which falls within OF2), the testing accuracy increases from 83.60% to 86.80%.

More channels were then added by progressively enlarging the dataset around the previously selected positions, as specified in the first column of Table 1. The results show that the accuracy increases with the number of channels if the corresponding fiber positions lie within the glued segments OF1 and OF2. Vice versa, the accuracy worsens when including fiber positions falling too close to or even outside the plate margins, as the prevailing effect in this case is the addition of environmental noise. The fiber channel selection resulting in the higher accuracy is highlighted in Table 1, with an overall testing accuracy of 91.20%. The corresponding confusion chart is shown in Figure 6. The chart reports the number of correct predictions along its diagonal, while the boxes off diagonal report the number of incorrect predictions. We see that the most frequent errors of the trained models occur between the “undamaged” class and D3, as well as between D3 and D4. Vice versa, the classes D1 and D2, corresponding to “defects” placed along a direction orthogonal to the glued fibers, give rise to the best results. It is important to remark that the sensor channels added progressively as precursors are not statistically independent, due to the limited spatial resolution of our sensor compared to the plate dimensions. Better results are expected adopting a distributed sensor with a higher spatial resolution, because in this case the SVM could be trained with a richer set of statistically independent channels. The defect D4 is detected less accurately than D3, despite the symmetry of our experiment. This can be explained by considering that the positions along OF1 and OF2 made to vibrate by the Lamb waves interacting with D4 are closer than those hit by the Lamb waves interacting with D3. Due to the finite spatial resolution of our sensor, the closer fiber channels are less informative, thus providing less accurate results in terms of damage detection. We also note that the signals related to the various channels are all available and acquired simultaneously; therefore, the use of more channels does not result in increased sensor complexity, but only in higher computational effort. As regards the latter aspect, the time required to train each SVM on a notebook computer with i7 core processor ranged from ≈ 0.5 s for a single channel to ≈ 6 s for 18 channels (row highlighted in Table 1).

The last experiments were performed with the aim of assessing the capability of the trained models to give a reasonable prediction, even when the defect lies in a position not included in the training dataset. To this end, we performed 10 more tests with the 5 g mass placed in position D3_1 and 10 more tests with the mass placed in position D4_1 (see Figure 7). The newly chosen positions are 75 cm distant from D3 and D4, respectively. The SVM classifiers trained with the combination of fiber channels highlighted in Table 1 (i.e., the one giving the best accuracy) were used to predict the damage positions, based on the ϕ-OTDR signals captured at the selected positions. The SVM providing the maximum score was then taken as the predicted class. The results are summarized in Table 2. We see that for the damage position D3_1, all 10 measurements led to a predicted class D3, which is the SVM class corresponding to the damage position closer to the one where damage occurred. For damage position D4_1, the situation is less clear, with four tests leading to a predicted class D4 (i.e., the one closest to the experimental conditions), while three tests gave D2 and another three tests gave D3. Despite the limited localization accuracy, it is worth remarking that the damage was always detected, i.e., all tests gave, as a prediction, one of the damage states of the plate. Still, the less accurate performance of defect D4_1 may be explained by considering the closest proximity between the OF1 and OF2 positions made to vibrate by the relevant Lamb waves.

## 5. Conclusions

In this work, a machine learning model based on support vector machines was presented for the classification of the conditions of a metallic plate using a piezoelectric actuator and an optical fiber. The latter was interrogated using a high-resolution homodyne ϕ-OTDR system. The results show that the proposed approach constitutes a promising approach for damage detection and localization, at least for a specific class of artificial damage (added mass). The measured accuracy was 91.20%, considering only five possible positions of the added mass. While further work is required to investigate the limitations of the proposed approach, especially in the case of a higher number of damage states, this exploratory study opens perspectives on a robust SHM technique using automatic classification of guided Lamb wave signals obtained with a distributed optical fiber sensor. Compared to other nondestructive techniques (NDTs) adopted in the SHM of civil and aerospace structures, such as radiographic testing or eddy current testing, the proposed approach can be applied in real time over the structure under operating conditions. Furthermore, the optical fiber sensor can be easily installed on the monitored structure compared to multiple PZTs or strain gauges. Regardless of the adopted sensing technology, it is generally recognized that machine learning and deep learning approaches are advantageous compared to traditional methods, as they only require feature extraction and classification [22]. A limitation of the proposed method is the high sensitivity of the phase-sensitive OTDR technology to ambient noise, which leads to the necessity of collecting a large dataset with the plate in both pristine and damaged conditions for robust SVM training. Furthermore, the interference fading, occurring when the phase is recovered at a point of destructive interference, may result in a very low SNR at some random fiber positions.

## Figures and Tables

**Figure 1 sensors-23-01099-f001:**
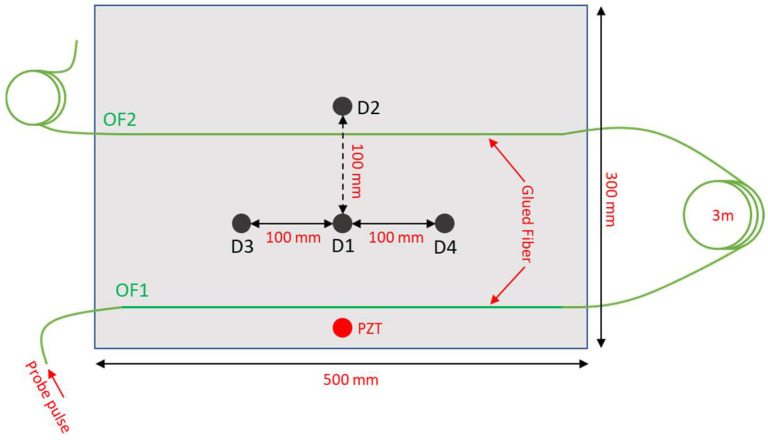
Aluminum plate used for the experiments. The red circle represents the PZT acting as the actuator, while the black circles represent the position of the 5 g lumped mass used to simulate the defect. The green line represents the path of the optical fiber sensor.

**Figure 2 sensors-23-01099-f002:**
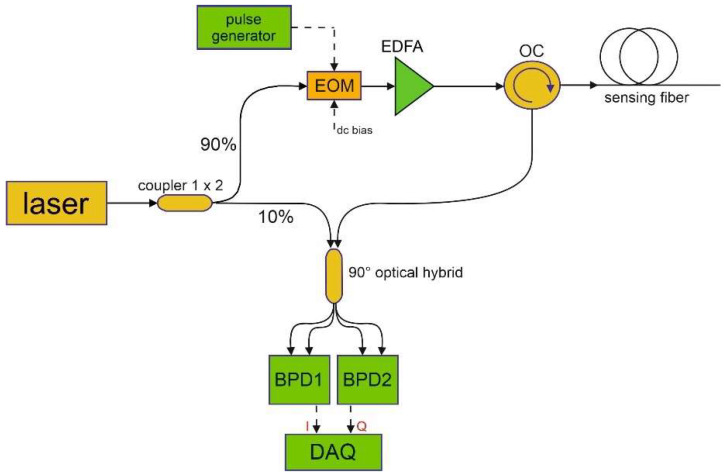
Experimental setup for ϕ-OTDR measurements (EOM: electro-optic modulator; EDFA: erbium-doped fiber amplifier; OC: optical circulator; BPD: balanced photodetector; DAQ: data acquisition).

**Figure 3 sensors-23-01099-f003:**
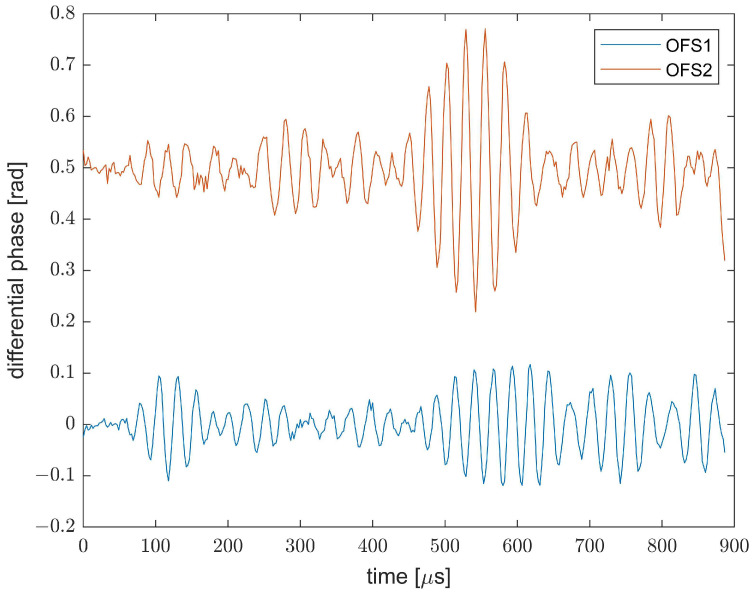
ϕ-OTDR signals acquired along the OF1 and OF2 fiber strands. The signal from OF2 has been vertically shifted by 0.3 rad for clarity purposes.

**Figure 4 sensors-23-01099-f004:**
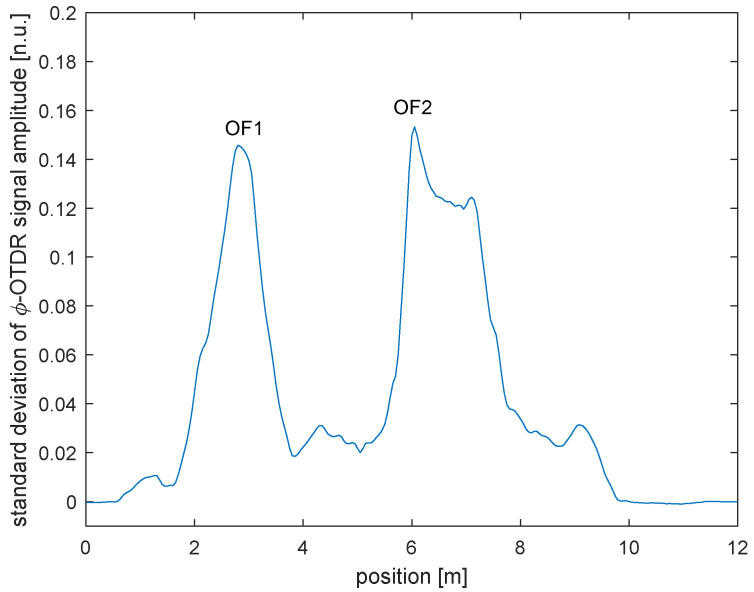
Standard deviation of the ϕ-OTDR signal amplitude as a function of the fiber position.

**Figure 5 sensors-23-01099-f005:**
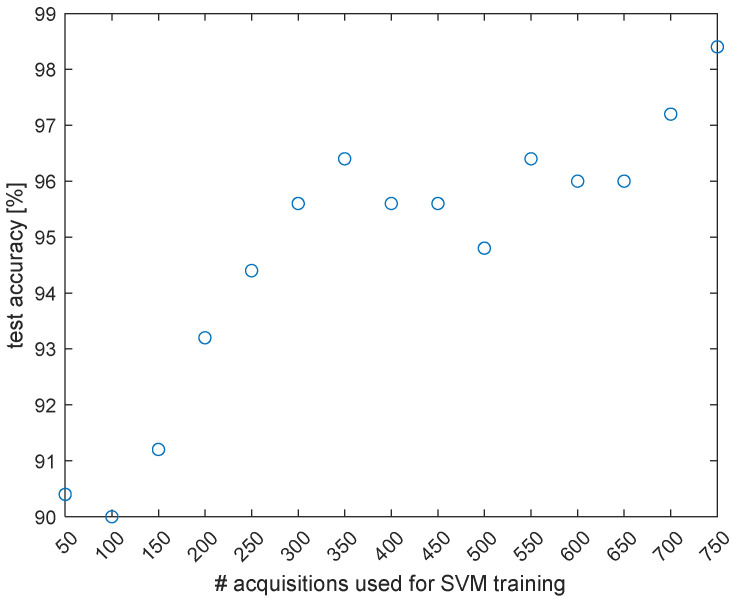
Accuracy of the SVM built for plate damage detection, as a function of the number of acquisitions used for training.

**Figure 6 sensors-23-01099-f006:**
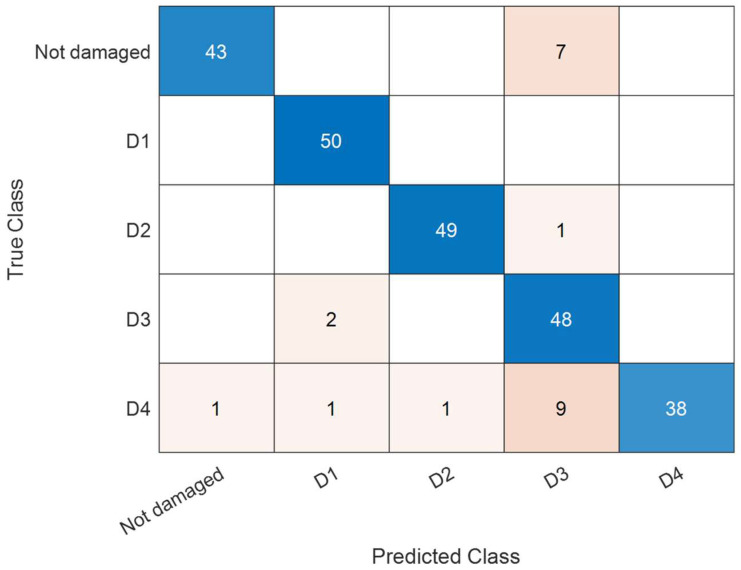
Confusion chart for the fiber channels highlighted in Table 1.

**Figure 7 sensors-23-01099-f007:**
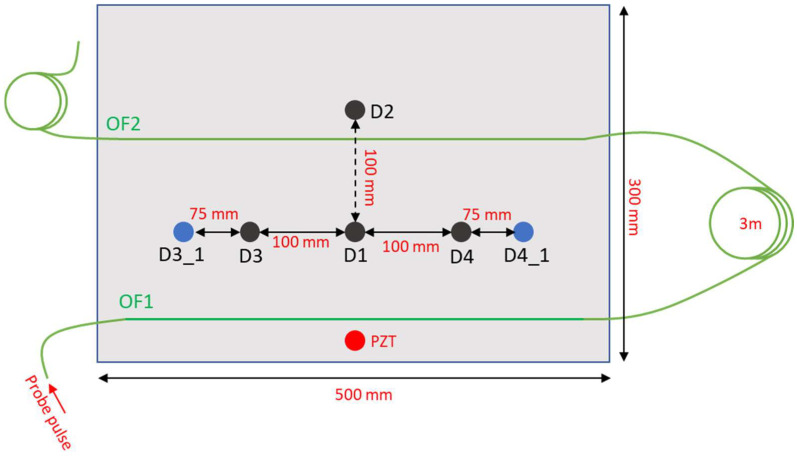
Aluminum plate used for the experiments, with the blue circles representing the positions used for testing but not for training.

**Table 1 sensors-23-01099-t001:** Testing accuracy of the SVM classifiers used for damage detection and localization, for various combinations of fiber channels.

Fiber Channels	Testing Accuracy [%]	Undamaged	D1	D2	D3	D4
z = 2.80 m	83.60	44/50	45/50	43/50	44/50	33/50
z = 2.80 m z = 6.05 m	86.80	46/50	48/50	48/50	48/50	27/50
z = 2.75:0.05:2.85 m z = 6.00:0.05:6.10 m	88.40	44/50	49/50	48/50	48/50	32/50
z = 2.70:0.05:2.90 m z = 5.95:0.05:6.15 m	90.00	42/50	49/50	49/50	50/50	35/50
z = 2.65:0.05:2.95 m z = 5.90:0.05:6.20 m	90.80	43/50	49/50	49/50	49/50	37/50
z = 2.60:0.05:3.00 m z = 5.85:0.05:6.25 m	91.20	43/50	50/50	49/50	48/50	38/50
z = 2.55:0.05:3.05 m z = 5.80:0.05:6.30 m	90.40	43/50	50/50	49/50	48/50	38/50
z = 2.50:0.05:3.10 m z = 5.75:0.05:6.35 m	90	41/50	50/50	49/50	48/50	37/50
z = 2.45:0.05:3.15 m z = 5.70:0.05:6.40 m	87.6	36/50	50/50	49/50	48/50	36/50

**Table 2 sensors-23-01099-t002:** Results of SVM testing with damage positions different from the ones used for training.

Real Damage Position	Not Damaged	D1	D2	D3	D4
D3_1	0	0	0	10	0
D4_1	0	0	3	3	4

## Data Availability

Data are available upon reasonable request.

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
