# Peer review of "Automated Damage Detection Using Lamb Wave-Based Phase-Sensitive OTDR and Support Vector Machines"

_sensors, 2023, doi:10.3390/s23031099_

Round 1

Reviewer 1 Report

The paper entitled "Automated damage detection using Lamb-wave-based phase-sensitive OTDR and Support Vector Machines" (by Zahoor et al.) deals with Lamb wave-based phase sensitive optical time domain reflectometer  testing and utilizing Machine Learning approach, is an original scientific work with promising technological applications in seismic risk mitigation. The paper deserves publication to the Sensors after the following remarks is taken into account.

Remark: The authors must discuss in the Discussion-Conclusions section extensively all possible limitations and shortcomings of the proposed method in comparison with other classical approaches in  detecting damage on buildings.

Author Response

We thank the Reviewer for his/her positive comments. The Conclusions section has been enriched with a discussion of the benefits and drawbacks of the proposed method.

Reviewer 2 Report

The reviewed manuscript might be considered as a further advancement of the results from Ref. 6. The authors attempt to adopt phase-sensitive OTDR technique together with SVM for damage detection and localization. The paper contains purely experimental results obtained for a certain experimental specimen and for a particular type of artificial damage.

The paper content is, in general, within the scope of the Sensors journal. However, certain improvements should be introduced into the manuscript prior to its acceptance. They are as follows:

1) In the introduction, it is mentioned that "This method [OTDR] can potentially replace conventional SHM sensors such as strain gauges and PZT transducers...". With this respect the authors could be encouraged either to discuss this statement in a more detailed way. For instance, the total price of a OTDR system could be compared over the conventional SHM approaches, and any other benefits different from those that "several sensors can be multiplexed along a unique optical fiber..." should be specified. Alternatively, this part of introduction could be reformulated in such way that would not cause any misunderstanding.

2) To increase the reproducibility of the obtained results, the authors are suggested to provide more details regarding all the equipment used in the experiment, i.e., specific type and manufacturers. Moreover, nothing is said about the size and shape of the artificial damage. How it was attached to the plate surface? Why it was essential to amplify the signal driving the PZT up to 400 V-p-p? Why central frequency of 38 kHz was chosen for the excitation signal?

3) Line 170: It is mentioned that "OTDR signals are usually contaminated by environmental noise". To what extent, the procedure described in lines 170-172 allows to address the issue with environmental noise? Were the external conditions during the experiment (i.e., temperature, humidity) somehow controlled?

4) Line 192: "a not perfect isolation of that fiber piece capturing a fraction of..." What does this mean? How to avoid this issue?

5) Damage locations D3 and D4 are symmetric against the vertical axis passing through the center of the test specimen. How could the authors explain sufficiently worse results obtained for the position D4 even in such ideal conditions?

6) In the Conclusion "an excellent capability of damage detection and ... good performance in terms of damage localization" are mentioned. These statements are sounding too optimistic. From the results presented in the paper, it might be only judged that the implemented OTDR-technique could detect the presence of particular type of artificial damage (however, with the probability around 90%) located in specific positions and hopefully could be used for its localization. To assure such statement, the authors should either perform sufficiently more comprehensive studies or formulate the Conclusion section in some other way.

Author Response

The reviewed manuscript might be considered as a further advancement of the results from Ref. 6. The authors attempt to adopt phase-sensitive OTDR technique together with SVM for damage detection and localization. The paper contains purely experimental results obtained for a certain experimental specimen and for a particular type of artificial damage.The paper content is, in general, within the scope of the Sensors journal. However, certain improvements should be introduced into the manuscript prior to its acceptance. They are as follows:

  • In the introduction, it is mentioned that "This method [OTDR] can potentially replace conventional SHM sensors such as strain gauges and PZT transducers...". With this respect the authors could be encouraged either to discuss this statement in a more detailed way. For instance, the total price of a OTDR system could be compared over the conventional SHM approaches, and any other benefits different from those that "several sensors can be multiplexed along a unique optical fiber..." should be specified. Alternatively, this part of introduction could be reformulated in such way that would not cause any misunderstanding.

The statement cited by the Reviewer has been clarified. In particular, the advantage of using optical fiber sensors such as FBGs instead of PZTs was already mentioned (low weigh, ease of installation, immunity to electromagnetic interference). The use of the phi-OTDR technology, which is a fully distributed sensing technique, brings the additional advantage of a potentially larger number of sensing points with respect to FBGs. However, this depends on the number of effective sensing points, which is simply the ratio between the length of the sensing fiber and the sensor’ spatial resolution. This aspect has been clarified in the introduction.

  • To increase the reproducibility of the obtained results, the authors are suggested to provide more details regarding all the equipment used in the experiment, i.e., specific type and manufacturers. Moreover, nothing is said about the size and shape of the artificial damage. How it was attached to the plate surface? Why it was essential to amplify the signal driving the PZT up to 400 V-p-p? Why central frequency of 38 kHz was chosen for the excitation signal?

The model and manufacturer of the main components used for experiments have been provided into the revised manuscript. Details about the artificial damage have been provided as well. Finally, in the revised paper it has been specified that, without amplification of the PZT driving signal, the resulting Lamb waves would be too weak to be detected by our sensor. More in detail, in our configuration the detected Lamb wave had an amplitude of 100 mrad (see Fig. 3), against an r.m.s. noise of ≈10 mrad. Without amplification, the resulting Lamb wave amplitude would be ≈100 times smaller (about 1 mrad), thus well below the r.m.s. noise. The 38-kHz central frequency was chosen as it is the first resonance of the PZT employed as actuator.

  • Line 170: It is mentioned that "OTDR signals are usually contaminated by environmental noise". To what extent, the procedure described in lines 170-172 allows to address the issue with environmental noise? Were the external conditions during the experiment (i.e., temperature, humidity) somehow controlled?

The procedure described in the paper, consisting in acquiring several times the Lamb wave signal under the same nominal plate conditions and using these data to train the SVM, helps in obtaining a robust model of the damage plate conditions (see Fig. 5). The experiments were performed under non-controlled laboratory conditions, with a mean temperature of 20 °C. Environmental noise is mostly due to laboratory instrumentation and ambient vibrations of the lab bench used during experiments (no optical table with active vibration isolation was employed). 

  • Line 192: "a not perfect isolation of that fiber piece capturing a fraction of..." What does this mean? How to avoid this issue?

In DAS experiments, the fiber could be isolated from ambient noise by putting the plate (with the fiber) inside an insulting (e.g., polystyrene) box, and/or placing the plate over a vibration-isolated optical bench. However, these solutions, while being viable options in a laboratory environment, cannot be simply transposed in a real (e.g., industrial) environment. Therefore, our experimental conditions are somewhat more representative of the real-field conditions.

  • Damage locations D3 and D4 are symmetric against the vertical axis passing through the center of the test specimen. How could the authors explain sufficiently worse results obtained for the position D4 even in such ideal conditions?

As correctly noted by the Reviewer, the results relative to D4 are worse than those relative to D3 (see Tables 1 and 2). This could be explained by considering that, the positions along OF1 and OF2 made to vibrate by the Lamb waves interacting with the defect D4, are closer (in terms of curvilinear abscissa along the fiber), than the positions along OF1 and OF2 hit by the Lamb waves interacting with D3. Due to finite spatial resolution of our sensor, the closest positions are somewhat less informative than the more distant ones. In other words, the contributions from OF1 and OF2 when detecting D4 provide less distinct information than those involved with the detection of D3. These considerations have been added in the revised manuscript (lines 269-274 and 301-303).

  • In the Conclusion "an excellent capability of damage detection and ... good performance in terms of damage localization" are mentioned. These statements are sounding too optimistic. From the results presented in the paper, it might be only judged that the implemented OTDR-technique could detect the presence of particular type of artificial damage (however, with the probability around 90%) located in specific positions and hopefully could be used for its localization. To assure such statement, the authors should either perform sufficiently more comprehensive studies or formulate the Conclusion section in some other way.

We agree with the Reviewer. The presented results are only preliminary, and more comprehensive studies are necessary to establish the validity and performance of the proposed method. The conclusions paragraph has been reformulated according to the Reviewer’s suggestion.

Round 2

Reviewer 2 Report

The authors have carefully responded to all the comments. The paper could be accepted.